# Halophilic Fungi—Features and Potential Applications

**DOI:** 10.3390/microorganisms13010175

**Published:** 2025-01-15

**Authors:** Lyudmila Yovchevska, Yana Gocheva, Galina Stoyancheva, Jeny Miteva-Staleva, Vladislava Dishliyska, Radoslav Abrashev, Tsvetomira Stamenova, Maria Angelova, Ekaterina Krumova

**Affiliations:** 1Departament of Mycology, The Stephan Angeloff Institute of Microbiology, Bulgarian Academy of Sciences, Acad. G. Bonchev Str. Bl.26, 1113 Sofia, Bulgaria; lyovchevska@microbio.bas.bg (L.Y.); j_m@microbio.bas.bg (J.M.-S.); dishliyskav@microbio.bas.bg (V.D.); rabrashev@microbio.bas.bg (R.A.); tsvetomirast@microbio.bas.bg (T.S.); mariange@microbio.bas.bg (M.A.); 2Departament of General Microbiology, The Stephan Angeloff Institute of Microbiology, Bulgarian Academy of Sciences, Acad. G. Bonchev Str. Bl.26, 1113 Sofia, Bulgaria; yanagocheva@microbio.bas.bg (Y.G.); galinadinkova@microbio.bas.bg (G.S.)

**Keywords:** filamentous fungi, saline environment, adaptation, application, ecology

## Abstract

Extremophiles are of significant scientific interest due to their unique adaptation to harsh environmental conditions and their potential for diverse biotechnological applications. Among these extremophiles, filamentous fungi adapted to high-salt environments represent a new and valuable source of enzymes, biomolecules, and biomaterials. While most studies on halophiles have focused on bacteria, reports on filamentous fungi remain limited. This review compiles information about salt-adapted fungi and details their distribution, adaptation mechanisms, and potential applications in various societal areas. Understanding the adaptive mechanisms of halophilic fungi not only sheds light on the biology of extremophilic fungi but also leads to promising biotechnological applications, including the development of salt-tolerant enzymes and strategies for bioremediation of saline habitats. To fully realize this potential, a comprehensive understanding of their ecology, diversity and physiology is crucial. In addition, understanding their survival mechanisms in saline environments is important for the development of astrobiology. The significant potential of applications of halophilic fungi is highlighted.

## 1. Introduction

Hypersaline environments occur worldwide, with some of the most extensively studied examples being the Dead Sea and the Great Salt Lake. Similar environments exist in countries such as Bulgaria, Australia, and Romania. Of note are the anoxic salt basins at the bottom of the Red Sea and the eastern Mediterranean, characterized by strong temperature, depth, and salinity gradients. Saline environments also occur in cold conditions, such as the salt channels in the polar ice [1]. Organisms inhabiting hypersaline environments are known as halophilic organisms. Halophilic microorganisms, which include bacteria, archaea and eukaryotes, represent an interesting class of extremophiles. They represent the natural microbial communities of hypersaline ecosystems and require sodium ions for their growth and metabolism [2]. These organisms are valuable sources of enzymes, biomolecules and biomaterials due to their unique properties, making them highly desirable for various biotechnological applications in extreme environments. Although thermophilic and alkaliphilic extremophiles are widely used in industrial applications, halophilic microorganisms are relatively underutilized despite their remarkable properties [1]. According to Oren [1], halophiles are defined as microorganisms that grow optimally in NaCl concentrations above 0.2 M [1]. Based on their salt tolerance, they are classified into three groups: weak halophiles (1–3% NaCl), moderate halophiles (3–15% NaCl), and extreme halophiles (15–30% NaCl). Microorganisms adapted to saline habitats that can grow and develop in both the presence and absence of salt are called halotolerants [3].

Tolerance parameters and salt requirements depend on temperature, pH and growth medium. Therefore, halophiles are adapted to and limited by certain environmental factors. These microorganisms, capable of optimally surviving and thriving under a variety of extreme environmental factors, are engineered polyextremophiles [2]. Halophilic fungi, as representatives of the eukaryotic halophiles, occur not only in hypersaline lakes but also in glacier ice because during freezing, dissolved substances are displaced into the liquid water between the crystals by the expansion of the ice crystals. Extremophilic or extremotolerant fungi have the necessary mechanisms to balance cellular osmotic pressure and ion concentration, stabilize cell membranes, and neutralize intracellular oxidative stress. They typically grow slowly as large amounts of energy are directed toward cellular mechanisms required to survive in adverse conditions. The polymorphism and meristematic growth of such fungi could be an adaptation to life under extreme conditions [4]. In addition, these microorganisms are adaptable to environments with high UV radiation and sometimes to extreme pH values [5].

The interest in halophilic filamentous fungi stems from the fact that, as polyextremophiles, they have developed successful survival and development strategies under numerous unfavorable environmental factors. The study and elucidation of the tolerance mechanisms of these eukaryotes would contribute to the understanding of the limits of life both on Earth and in the universe, thanks to their remarkable tolerance properties and their adaptability in terms of ecology and morphology, which is one of the essential reasons to study their relationship to exobiology [6].

Microorganisms adapted to saline environments have many actual or potential applications in various areas of social life. To date, biotechnological applications of halophilic microorganisms have been mainly associated with the use of halophilic bacteria, while the broad potential of halophilic fungi is still poorly understood and therefore rarely used in practice [7]. Continually increasing antibiotic resistance determines the constant need for new antibiotics. Some of the halophilic fungi isolated so far show high antibacterial activity, making them attractive as potential antibiotic producers [8]. Halophilic and halotolerant microorganisms are considered reliable sources of antitumor metabolites with fewer side effects [2].

The bioremediation potential of halotolerant and halophilic fungi to decolorize synthetic dyes has been demonstrated [9].

In this review, we have collected and summarized information on halophilic fungi from recent studies. It discusses their biodiversity, their adaptation mechanisms to adverse environmental conditions and their enormous potential for application in various areas of social life.

## 2. Biodiversity and Taxonomy of Fungi in Saline Habitats

For a long time, places with extreme living conditions were considered uninhabitable. Since intensive research into these habitats began, a large number of extremophilic microorganisms have been discovered [10]. The mycobiota inhabiting natural saline environments consists of halotolerant and halophilic fungi, represented not only by previously known species but also by new and rare ones. Several studies have examined fungal diversity in different saline environments. According to the authors, halophilic and halotolerant fungi are found in the Dead Sea [11,12], desert soils [13], estuaries and salt lakes [14], mangroves [15] and solar salt marshes on various continents [5,15,16].

The majority of reported halophilic fungi belong to the division *Ascomycota*, but representatives of the division *Basidiomycota* have also been reported. Among the *Ascomycota* representatives, *Capnodiales*, *Dothideales*, and *Eurotiales* are the most important halophilic genera. Representatives of the species *Aspergillus sydowii*, *Aspergillus versicolor*, *Aureobasidium pullulans*, *Hortaea werneckii*, *Penicillium chrysogenum*, *Phaetotheca triangularis* and *Trimmatostroma salinum* have been described as halophiles. Members of the orders *Trichonosporales*, *Sporidiales*, and *Wallemiales*, including the halophiles *Wallemia ichthyophaga*, *Wallemia sebi*, and *Wallemia muriae*, are derived from the division *Basidiomycota* [17,18,19]. The most dominant is the *Penicillium* genus. Many different *Penicillium* species have been reported from saline habitats (hypersaline area, saline soil, saline marine waters, lakes). For the first time, *Aspergillus* and *Penicillium* were found to be dominant, while *Cladosporium* was less common in the Dead Sea. Halophilic fungal isolates from man-made solar salts in Pattani Province, Thailand, belong to the genera *Aspergillus*, *Cladosporium*, *Curvularia*, *Diaporthe*, *Ectophoma*, *Fusarium* and *Penicillium* [15,20]. Variations in their growth at different salt concentrations have been noted. Some *Aspergillus* species are facultative halophiles, while *Cladosporium* and *Penicillium* species are exclusively facultative halophiles, and *Aspergillus penicillioides* and *Aspergillus unguis* are considered obligate halophiles [21]. The identification of halophilic fungi from different geographical regions proves the presence of fungi of the genera *Aspergillus*, *Penicillium*, *Alternaria* and *Cladosporium* [14]. Many of them show xerotolerance and grow well in low aw conditions.

Furthermore, it has been found that halophilic fungi are mainly presented of melanized fungi belonging to the genera Cladosporium, Aspergillus, Eurotium [22], *Hortaea werneckii* [23] and *Stachybotrys* [24,25]. Accumulated melanin in their cells is thought to protect them from harsh conditions [26]. The melanized structure of the cell wall prevents glycerol decreasing by reducing the pore size [27]. It is an interesting fact that the extremely halotolerant fungus *Hortaea werneckii* can form an association with the alga *Dunaliella atacamensis* and with *D. salina*, but the mutual benefits for the partners are not yet fully understood [3]. Leita˜o and Enguita [28] reported a halophilic endophilic *Penicillium* sp. as a gibberellin producer. Abiotic stress conditions such as salinity are known to negatively impact plants, and the presence of bioactive gibberellin is a crucial factor for plant growth under such conditions. It was reported that a new halophilic fungus, *Penicillium imranianum*, was isolated from an artificial solar salt marsh in Phetchaburi Province in Thailand. The authors claim that it is the halophilic *Penicillium* [16].

From the studies of other authors as obligate halophiles *Gymnascella marismortui* [11], *Wallemia ichthyophaga* [29,30], *Trichosporium* sp. [31], *Aspergillus penicillioides* and *Aspergillus unguis* [32] are reported. Species of the genera *Aspergillus* and *Penicillium* are predominant in the mangrove salt marshes of Goa, India. They also have high resistance to heavy metal ions such as Pb^2+^, Cu^2+^ and Cd^2+^. The Dead Sea is an extremely hypersaline environment with a concentration of about 340‰ of total dissolved salts and a content of divalent ions such as magnesium and calcium that is higher than that of monovalent ions. *Gymnascella marismortuispecies*, *Penicillium westlingii* and *Ulocladium chlamydosporum* were initially reported as representatives of the local mycobiota [11]. Later, information on representatives of *Aspergillus* and *Cladosporium* spp. as well as *Penicillium westlingii* and *G. marismortui* [32,33,34] is outlined. Table 1 presents the biodiversity of some salt-adapted fungi.

## 3. Halophilic Fungi and Their Adaptation Mechanisms to High Salt Concentrations

The survival of halophilic fungi at high salinities is facilitated by a range of adaptation mechanisms at various levels: cellular (e.g., increased cell wall thickness and specialized transporter systems), genetic (e.g., transport-related genes and high acidic amino acid content), enzymatic, and metabolic pathways (e.g., oxidoreductase, superoxide dismutase, and the high osmolarity glycerol signaling pathway) (Figure 1). The membrane structure of halophilic microorganisms plays an important role in adapting to saline conditions. It protects the cell from the harmful effects of elevated salt concentrations and maintains osmotic homeostasis in the cell through appropriate membrane fluidity [45]. The cell wall and cell membrane of fungal cells are the first to be exposed to the increased salinity of the environment. Fungi include structural changes to the cell wall and cell membrane as well as the presence of pigments and/or hydrophobins in response to increased salinity [46,47,48]. Structural changes in response to increased salt in the medium include thickening of the cell wall and a change in lipid composition [49]. The presence of pigments (melanins and carotenoids) that protect halophilic fungi from UV rays and the harmful effects of sunlight is another example of its adaptation [50,51,52]. For example, to defend themselves against environmental stress, black yeasts, which are halophilic fungi, produce melanin [53]. As the essential adaptation for fungal survival at high salinities, the presence of melanin in the cell wall inhibits water loss and leakage of intracellular compatible solutions and helps to maintain high membrane fluidity [8,47]. The cell can adhere to hydrophobic surfaces with the help of hydrophobins and overcome the water–air interface [54]. As reported by Fernando et al. [46], these proteins strengthen and stiffen the cell wall to promote haloadaptation.

Other important structural changes in the cell wall include the biosynthesis of crystalline chitin and the introduction of α-glucan branch points, which help halophilic microorganisms to improve the impermeability of the cell wall and to protect themselves from environmental stresses [55].

In many natural hypersaline environments, the concentrations of toxic Na^+^ ions are significantly higher than those of K+ ions, so maintaining a stable and high intracellular K^+^/Na^+^ ratio is essential for survival under such conditions. To maintain cellular balance, filamentous fungi have developed an adaptive mechanism. *W. ichthyophaga* and *H. werneckii* are able to maintain high K^+^/Na^+^ H ratios over a wide range of environmental Na^+^ concentrations [56]. *H.werneckii* has a diverse and highly enriched population of cation transporters, presumably to allow it to rapidly adapt to the ever-changing NaCl (and other salt) concentrations characteristic of its natural environment. This regulated transport of K^+^ and Na^+^ allows *H. werneckii* to maintain homeostasis. By maintaining a low sterol-to-phospholipid ratio and reducing fatty acid cycle length and phospholipid saturation, *H. werneckii* maintains membrane fluidity over a wide range of salinities [56]. *W. ichthyophaga* also regulates the influx and efflux of cations, but prevents their entry primarily through dynamic restructuring of the cell wall. All of this ensures the survival and prosperity of *W. ichthyophaga* under conditions of consistently high salinity [56].

The survival of halophilic fungi under conditions of increased environmental salinity also requires a number of adaptations at the genetic level, such as a rapid response to water loss. In *H. werneckii* and *W. ichthyophaga,* this is achieved through the synthesis of glycerol and several other compatible solutions. Adaptation at the genetic level is associated with the expression of the key enzyme of glycerol synthesis, Gpd1, which is under the control of the high-osmolarity glycerol (HOG) signaling pathway. This signaling pathway is crucial for adaptation to high osmolarity environments. Plemenitaš et al. [47,56] compared HOG signaling in *H. werneckii* and *W. ichthyophaga* [57]. In *H. werneckii*, HwHog1 kinase regulates the differential induction or repression of osmoactive genes depending on osmolarity, and this is achieved through physical interactions with chromatin and RNA polymerase II [58]. Upon activation, HwHog1 kinase translocates to the nucleus, where it binds to the chromatin of osmoactive genes and induces or represses their expression [58].

The extremely halotolerant black yeast *H. werneckii* possesses 95 osmoresponsive genes. More than half of them are related to fungal metabolism and energy. Thirteen unclassified SOL genes represent a specific transcriptional response unique to *H. werneckii*. In total, 36 genes are Hog1-dependent genes related to long-term adaptation to extreme environments. This set of osmoresponsive genes is responsible for the hypersaline adaptation of extremely halotolerant eukaryotes [57].

Comparing media with moderate (3 M NaCl) and extreme (4.5 M) osmolarity, a transcriptional response to hyperosmolar stress involving 95 differentially expressed genes was reported. Chromatin immunoprecipitation (ChIP) data showed that 36 of these genes physically interacted with HwHog1 kinase during long-term adaptation to extreme conditions [58]. Colocalization with RNA polymerase II was observed for 17 of these 36 genes. More than half of the differentially expressed genes are related to general metabolism and energy production, and the remaining eight active genes are involved in mitochondrial biogenesis, protein biosynthesis, protein quality control, transport facilitation, cell cycle, and cell wall processes [58]. In *W. ichthyophaga,* there are only two isoforms of Hog1 kinase. Under optimal osmotic conditions, WiHog1 kinase is constitutively phosphorylated, while under hyperosmotic or hypoosmotic shock, it is dephosphorylated [59]. Constitutive phosphorylation of Hog1-like kinase in *W. ichthyophaga* may help maintain its obligate halophilic nature. Maintenance of ionic balance in the halophilic fungus *Aspergillus montevidensis* ZYD4 is primarily mediated by differential expression of genes related to the MscS protein and increased antioxidant activity. *A. montevidensis* ZYD4 increases alanine, aspartate, and glutamate metabolism as well as the ornithine cycle to accumulate various amino acids and amino acid derivatives (e.g., alanine, ornithine, proline, aspartic acid, and urea) in response to high salinity conditions [60]. Two genes encode the ENA-ATPases DhENA1 and DhENA2, which are involved in salt extrusion in *D. hansenii*. The former gene is highly expressed in the presence of higher salt concentrations, while the expression of the latter gene depends on high pH [51].

A notable feature of *H. werneckii* is the significant genetic redundancy, most likely resulting from whole genome duplication. According to Gostinčar et al. [17], such redundancy can serve as a reservoir of genetic variability, which is valuable in stressful environments that require strong adaptability [17]. The halophilic fungus *Aspergillus montevidensis* ZYD4 not only accumulates various soluble sugars in response to elevated salt conditions (e.g., maltotritol, D-glucose, heptose and trehalose-6-phosphate), but also activates the hexose monophosphate (HMP) pathway and initiates the *Tricarboxylic acid* (TCA) cycle via the acetyl-CoA reaction of fatty acid β-oxidation, which leads to increased production of oxaloacetic acid, leading to a series of intermediates and NAD(P)H (or FADH2). In addition, the osmotically adapted fungi significantly increase lipid transport and metabolism (β-oxidation of fatty acids and synthesis of saturated fatty acids). The regulation of the NADH pool is also linked to the activity of the electron transport chain to generate sufficient ATP for salt adaptation. The addition of metabolites such as neohesperidin, urea, aspartic acid, alanine, proline, and ornithine significantly stimulated the growth (*p* ≤ 0.05) and morphological adaptations of *A. montevidensis* ZYD4 grown in hypersaline medium [60].

It should be emphasized the great potential genetic engineering for planning and building synthetic gene networks, new metabolic pathways or control circuits, which can be created fungi with improved desired properties [61]. Genetically modifying fungi to increase secondary metabolite production requires manipulation of biosynthetic gene clusters (BGCs) that encode essential enzymes and regulatory proteins. Techniques such as CRISPR/Cas9 enable the precise activation or suppression of specific genes within these clusters [62]. The application of CRISPR/Cas9 enables precise modifications of genes involved in enzymatic biosynthetic pathways. This has led to higher enzyme yields and improved enzyme properties such as stability and activity under industrial conditions [63]. For example, the industrial fungus Trichoderma reesei has been genetically engineered to produce significant amounts of proteins, including egg white and milk protein. CRISPR-Cas9 was used to delete the cre1 gene, a transcriptional repressor of cellulase production [64]. Deletion of cre1 from the Trichoderma reesei genome resulted in a significant increase in cellulase production by alleviating the repression of genes associated with cellulase activity.

Furthermore, the discovery of neutral loci for gene integration in fungi such as *Aspergillus oryzae* could enable more stable and predictable expression of manipulated traits, thus expanding the potential uses of fungi in biotechnology and agriculture [65]. These genetic changes in fungi that produce metabolites of biotechnological importance are expected to increase the yield of the desired metabolite and promote its effective use in biotechnological applications.

Adaptation to a high salinity environment by maintaining the osmotic balance of the ionic cytoplasm and stabilizing the protein machinery makes these halophilic fungi attractive for industrial enzyme processing [3]. At the biochemical and metabolic level, fungi adapted to environments with low water content (low aw) accumulate organic compounds in the cytoplasm, i.e., osmoprotectors (so-called compatible solutions), as a survival strategy [60,66]. This strategy is known as the “organic osmolyte” mechanism [6]. They can arise through synthesis or accumulation in the environment. The most common compatible solutions are neutral or zwitterionic and include amino acids, sugars, polyols, betaines and ectoine, as well as derivatives of some of these compounds. Their main function is to stabilize intracellular Na^+^ at non-toxic levels. The HOG pathway is an important signaling mechanism in fungi for controlling cellular stress responses, which has been specifically studied in *Saccharomyces cerevisiae* [67]. Activation of this pathway leads to glycerol production, which primarily contributes to the restoration of cellular osmotic balance [68]. Using the halotolerant black yeast *H. werneckii* and the halophilic fungus *W. ichthyophaga*, Plemenitas (2021) showed that although both fungi use the HOG signal transduction pathway to adapt to external salinity, their activation mechanisms of this pathway is different [69]. *W. ichthyophaga* produces extracellular polysaccharides that protect cells under conditions of increased salt concentration and also in drought [70].

Another crucial challenge for microorganisms in saline habitats is oxidative stress. Stressful conditions disrupt the electron transport chains in mitochondria, leading to reverse electron flow and undesirable oxidation of oxygen by complex I and formation of reactive oxygen species (ROS), respectively [71]. Some studies in plants suggest that the acquisition of salt tolerance may be due to improved resistance to oxidative stress [50]. Studies on the response of *H. werneckii* to oxidants show that its ability to degrade hydrogen peroxide over a wide range of salinities is as high or even higher than that of *S. cerevisiae*, which is stressed by exposure to 3% salt [53]. Furthermore, the molecular chaperones Hsp70 and Hsp90 of *H. werneckii* are upregulated under high-salt conditions and contribute to the control of proteins damaged by stressful conditions [58].

The fungal adaptations to high salinity are summarized in Table 2.

## 4. Applications of Halophilic Filamentous Fungi

Salt-adapted fungi represent a rich genetic resource that holds promise for the development of robust industrial organisms. This review highlights their considerable prospects. Due to the intriguing biomolecules they contain, halophilic fungi have garnered a lot of attention lately for industrial use.

### 4.1. In Biotechnology

The discovery of extremophilic fungi has played a crucial role in the advancement of biological and biotechnological research [71]. In recent years, the biotech industry has experienced significant growth as it seeks environmentally friendly alternatives to chemical production. However, biologically derived products such as biofuels, biochemicals and bioplastics often have higher production costs than their chemical counterparts. This cost difference may be due to a combination of factors, including increasing costs of raw materials such as starch-derived glucose, the ever-increasing demand for fresh water exacerbating water scarcity concerns, fermentation disruptions due to environmental pollution, and reduced product efficiency significant energy costs associated with sterilizing fermenters and media.

Halophilic and halotolerant fungi are valuable sources of enzymes with potential use in biotechnology, bioremediation, medicine, biofuel industry, etc. *A. gracilis* TISTR 3638 α-amylase isolated from a solar salt plant shows superior activity under conditions compared to other fungi commercial amylases with high salt content when used for wastewater remediation. This property makes it particularly suitable for the treatment of industrial wastewater contaminated with metal ions [7]. A novel halophilic extracellular lipase isolated and characterized from *Fusarium solani* was reported by Geoffry et al. [72].

This enzyme exhibits both hydrolytic and synthetic activities and could be extremely useful for various biotechnological applications.

The halophilic fungus *Stachybotrys microspora* produces a halo-alkali-tolerant endoglucanase named EG1 [45]. This enzyme was proven to be active even in the presence of 5 M NaCl. Additionally, it remains active in the presence of various surfactants and detergents. All the characteristics listed so far give us reason to believe that this enzyme is suitable for use in the textile industry and the digestion of cellulose waste. Interestingly, another cellulase (EG2) with lower halotolerant properties and optimal activity at 0.85 M NaCl was purified from the same fungal strain [45].

Halophilic fungi play an important role in industrial food fermentations as sources of various enzymes. For example, traditional soy sauce production primarily relies on the enzyme glutamyl transpeptidase, with the process occurring in high salt concentrations. During fermentation, the enzyme contributes to improving the taste of the soy sauce. Recently, Gamma-glutamyl transpeptidase from *Aspergillus sydowii* was obtained. The enzyme shows high activity at 18% NaCl and high thermostability, making it a promising candidate for the food industry [73].

Microbial proteases and lipases play an important role in an array of industries, from food processing to waste management. These processes often occur in the presence of high salt concentrations, which makes the search for new halophilic proteases crucial, due to their remarkable capabilities. The halophilic fungus *Penicillium camemberti* secretes proteolytic and lipolytic enzymes that are used for the production of Camembert cheese [45]. They also produce other halophilic enzymes such as esterases and lipases, which are widely used in the food industry. Furthermore, halophilic xylanases from these fungi have potential applications in savory food and seafood processing [46].

A halophilic fungal strain, *Aspergillus reticulatus* SK1-1, isolated from a saltern in the Republic of Korea produces an extracellular alkaline serine protease with maximum activity at 40–50 °C, pH 9.5–10.5, while retaining up to 69% activity at 7% NaCl. The potential of halophilic *A. reticulatus* proteases for diverse biotechnological applications, including food processing, detergents, textiles, and waste treatment, was highlighted [37].

A novel enzyme, produced by *Trichoderma asperellum* ND-1 and exhibiting halophilic characteristics, was successfully expressed in *Pichia pastoris*. This halophilic xylanase, designated as TaXYL10, demonstrates activity in high-salinity environments (up to 4.28 M NaCl) and in the presence of 10% ethanol. It retains optimal performance at a pH of 6.0 and a temperature of 55 °C, showing considerable stability within the pH range of 4.0 to 6.0. The enzyme exhibits a marked preference for beechwood xylan and effectively hydrolyzes xylan substrates, yielding xylotriose and xylobiose. Its properties suggest significant potential for industrial applications, especially in the production of xyloligosaccharides from agricultural waste under halophilic conditions [74].

### 4.2. In Bioremediation

In recent decades, heavy metal pollution has caused severe damage to the ecosystem and human health due to industrialization and technological progress [47]. Bioremediation is an alternative to physical and chemical methods for the separation and recovery of heavy metal ions from the polluted environment. The use of living organisms (algae, bacteria, fungi or plants) is an innovative method to reduce and/or recycle heavy metal pollution in a less dangerous form [48]. Microbial remediation is an environmentally friendly and cost-effective method in which microorganisms absorb, precipitate, oxidize and reduce heavy metals found in soil. It is particularly valuable in oil spill cleanup, wastewater treatment, and soil decontamination [49]. The remediation of extremely saline environments is becoming increasingly important and is closely related to the control of contamination of salt marshes and saline wastewater generated by various industrial activities [51]. Numerous studies have been carried out to reveal the involvement of halophilic bacteria and fungi in microbial remediation processes.

Bioremediation uses affordable, environmentally friendly and harmless approaches such as biodegradation, remediation or decolourisation using microorganisms [75]. In the industrial sector, bacteria, fungi and algae are used for biological wastewater treatment. They have high cost-effectiveness, easy-to-use operating systems, energy efficiency, high biomass volume, and the ability to produce value-added products that can be used for power generation and other applications [76,77].

Filamentous fungi have a number of advantages over bacteria and algae with regard to wastewater bioremediation. They produce a number of enzymes and surface proteins that play an important role during the biodegradation or biosorption of wastewater contaminants [76]. As heterotrophs, they can grow on multiple substrates under different conditions. Filamentous fungi have great potential for industrial wastewater treatment [78].

Wastewater treatment and bioremediation processes can benefit from the production of various hydrolases and oxidoreductases that are resistant to the presence of salt and have low water activity. Fungi from saline environments are also a source of numerous ligninolytic enzymes that could be useful in biomass conversion processes using poorly soluble lignin materials [52]. There are many studies in the literature on this possibility of halophilic fungi involved in the bioremediation of water and soil contaminated with heavy metals. Recently, it was reported that the halophilic fungus *Aspergillus* sp. and *Sterigmatomyces* sp. showed moderate to high adsorption of heavy metals and could be used for biosorption of cadmium, copper, iron, manganese, lead and zinc. Bano et al. [79] reported *A. flavus* and *S. halophilus* as very good adsorbents for heavy metals, with Fe and Zn being best removed from the liquid medium with an average of 85 and 84%, respectively [79].

The first attempt to use amylase to treat wastewater was performed using the halophilic species *Aspergillusgracilis*. The amylase enzyme was involved in this process. The mechanism involved in changing the dissolved oxygen value in wastewater was monitored using a DO meter. The DHN-melanin inhibitor, tricyclazole, inhibited the melanin in the halophilic fungi *H. werneckii* [7].

Halophilic fungi such as *Aspergillus gracillis*, *A. restrictus* and *A. pencilloides* mostly act based on the biosorption mechanism. Sterigomatomycetes are used in the removal of cadmium, copper, ferrous, lead and zinc [79]. In xenobiotic mycoremediation of wastewater, both *A. sydovii* and *A. destruens* were used and the study was evaluated using the GC_MS technique [80].

The bioremediation potential of halotolerant and halophilic fungi to decolorize synthetic dyes has been demonstrated and is being further investigated. *Aspergillus flavipes* MA-25, an isolated halotolerant fungus, can decolorize Reactive Black 5, a textile dye. Complete removal of the initial dye concentration of 0.2 g/L was achieved after 80 h of incubation. The primary method by which this fungus removed dye was bioadsorption [81]. A study by Iscen et al. [9] revealed that the hypersaline medium isolate *A. niger* removed 98.97% of Remazol Black B dye at a concentration of 100 ppm after two days of incubation. The authors reported the potential of halophilic fungi in the decolorization of dyes and salts in industrial wastewater [9]. A novel halotolerant fungal strain isolated by Sharma et al. [82] from rotten wood samples was identified as *Trametes flavida* WTFP2 [82]. It has the ability to effectively decolorize the harmful diazo dye Congo Red at a concentration of 100 mg/L within 24 h, achieving an impressive decolorization rate of 87.28%. Furthermore, at a higher dosage of 1000 mg/L, it showed a degradation rate of 70.19% within 72 h of incubation [82].

Xenobiotics are very persistent in the environment and contaminate water and soil [83]. González-Abradelo et al. [80] demonstrated for the first time the value of halophilic fungi in xenobiotic myco-remediation in high salinity conditions [80]. *Aspergillus sydowii* and *Aspergillus destruens* can eliminate over 90% of polycyclic aromatic hydrocarbons (PAHs) and pharmaceutical compounds (PhCs) under saline conditions. The fungal strains investigated used different mechanisms of action. *A. sydowii* used the mechanism of biodegradation and *A. destruens* used bioadsorption. Both xenobiotics are present in many saline wastewaters [80].

Phenol, a hazardous organic compound, and its derivatives are among the major groups of pollutants in industrial wastewater and pose a significant threat to the environment and human health. Classic approaches to phenol degradation involve chemical processes that can be costly and environmentally harmful [84]. Halophilic fungi tolerate high salt concentrations and are therefore suitable for the treatment of phenolic contamination in saline environments. In recent decades, intensive studies have been carried out to isolate and screen halophilic fungi that can utilize phenol as a sole carbon source. Efficient removal of phenol was demonstrated by the halotolerant fungus *P. chrysogenum* CLONA2 [85]. It has been suggested that *Debaryomyces* sp. cells immobilized in alginate beads degrade up to 99% of the phenol within 80 h without accumulation of intermediate products. Alginate beads containing live fungal cells remain active and effective and can be used in multiple cycles [86].

Fungi of the genera *Aspergillus*, *Pencilllium* and *Fusarium*, which can grow and degrade phenol and phenolic derivatives as sole carbon sources, have been isolated from sediments along the Suez Gulf and mangroves along the coasts of the Red Sea (Egypt). It has been demonstrated that the fungus *F. flavipes* has the highest efficiency in phenol degradation [84].

After 10 days of incubation in non-amended wastewater, *C. matritensis* and *A. fumigatus* isolated from olive brine wastewater were able to remove phenolics by 52.3 and 82.3%, respectively, by producing extracellular phenol oxidases. These results may indicate the potential application of these strains for the remediation of other saline wastewaters with high phenolic content, such as pickling and tannery wastewater [87]. The halophiles *A. sydowii* EXF-12860 and *A. destruens* EXF-10411 have shown excellent results in removing 100% of xenobiotics—PAHs and pharmaceutical compounds (PhCs)—in wastewater under saline conditions (>1 M NaCl), so they could be used for the downstream biotechnological treatment of various industrial wastewaters [80].

### 4.3. Biofuel Production

Due to increasing global energy consumption and the recent increase in global oil prices, biofuel sectors are receiving renewed interest [88]. At the beginning of the third millennium, fossil fuel prices began to rise sharply, and biofuels gained popularity due to growing global awareness of pollution and global warming, as well as the desire for self-sufficiency [89]. A variety of fuels, including solid biomass, liquid fuels and biogases, are considered biofuels because they are derived from biomass [90]. The biofuel industry has attracted significant attention and scientific interest due to increasing global energy consumption and recent increases in oil prices. As society becomes more aware of the environmental impact of fossil fuels, biofuels have emerged as viable and sustainable alternatives. The demand for biofuels has increased significantly due to increasing global energy consumption and the rising cost of traditional fossil fuels [69]. Biofuels include a variety of fuels including solid biomass, liquid fuels and biogas, which are all derived from biomass sources. These biofuels serve as an environmentally responsible means of energy production and offer a more sustainable and environmentally friendly alternative to fossil fuels [90]. The ability to break down lignocellulose into fermentable sugars is crucial for bioethanol production. In a previous study, Bano et al. [79] purified and characterized cellulase enzymes that have high activity in the presence of NaCl (150 g/L) [79]. A halotolerant enzyme was produced by *A. flavus* (TISTR 3637), an obligate halophilic fungus isolated from an artificial solar salt plant in Thailand. Halotolerant fungi, such *A. flavus* and *A. penicillioides*, are ideal for pretreatment and saccharification of biomass in saltwater environments because they produce cellulases that remain highly active even at high salt concentrations [35]. According to Molitoris et al. [33], several halophilic marine fungi (*Ascocratera manglicola*, *Astrosphaeriella striatispora*, *Cryptovalsa halosarceicola*, *Linocarpon bipolaris* and *Rhizophila marina*) show a significant ability to solubilize lignin [33]. A marine fungal strain *A. sydowii* BTMFS 55 showed extracellular β-glucosidase activity in different NaCl concentrations and could have potential applications in the production of bioethanol [91].

Halotolerant fungi offer a unique and promising solution to many of the challenges facing the biofuel industry. The adaptability of halotolerant and halophilic filamentous fungi opens a wide range of applications to improve the sustainability and profitability of biofuel production using environmentally friendly technologies [92]. For higher bioethanol yields, further research to optimize growth conditions and enzyme production is required and the genetic engineering of halotolerant fungi and their use in large-scale industrial processes must be researched.

### 4.4. In Medicine

Bioactive molecules produced by organisms are suitable anticancer drugs. Although the originally established natural substances against cancer are obtained from plant cells, microorganisms are an excellent alternative. They have many advantages that make them suitable producers: the large microbial diversity, their much easier manipulation and the possibility of physiological screening to discover new bioactive molecules with antitumor activity. In the search for new biomolecules, scientists are increasingly turning their attention to extremophiles. Halophilic and halotolerant microorganisms are considered reliable sources of antitumor metabolites with fewer side effects [8]. Several studies reported that fungi isolated from deep-sea sediments produce anticancer drugs. For example, the fungus *Diaporthe longicolla* FS429 produces longihalazin B, which shows antiproliferative activity against glioblastoma cells (SF-268) [40]. *Cladosporium cladosporiodes* HDN14-342 produces polyketides, clindanones A and B, and cladosporols F and G, which exhibit cytotoxic activity against human cervical, leukemia, and colon cancer cells [93]. *Phomopsis lithocarpus* FS508 isolated from sediment produces Tenelon H (benzophenone aldehyde) with cytotoxic activity against liver and lung cancer cell lines [41]. Cytotoxic activity against cervical cancer cells is demonstrated by penifelan D isolated from the deep-sea sediment *Penicillium fellutanum* HDN14-323 [42]. *Acaromyces ingoilii* FS121 collected from deep-sea sediments produced the cytotoxic agent acaromycin A, which showed cytotoxic activity against human breast (MCF-7), brain (SF-268), liver (HepG-2), and lung (NCI-H460) cancer cells [94]. The DFFSCS02 Engyodontiumone H obtained from *Engyodontium album* showed cytotoxic activity against the human histiocytic lymphoma U937 cells [43]. In addition, 4,8,10-trihydroxy-1,2,11,12-tetrahydroperylene-3-quinone, also called altertoxin VII, 92 (a polyketide isolated from *Alternaria* sp. SCIO41014), shows cytotoxic activity against human erythroleukemia [95] and 24-hydroxy-trichodimerol isolated from *Trichoderma reesei* has cytotoxic effects on cancer cells in the lung, breast and liver [96].

Many of the pigments produced by fungi also possess anticancer activity and low cytotoxicity to normal cells [97]. Thiodiketopiperazine alkaloids, i.e., Eutypellazine A-S, were isolated from the marine fungus *Eutypella* sp. MCCC. Among them, eutypellazines A-L showed anti-HIV activity [38]. The Antarctic sediment fungus *Penicillium granulatum* MCCC 3A00475 produces spirograterpene A that exhibits antiallergic activity against immunoglobulin E-mediated rat RBL-2H3 mast cells [44].

### 4.5. In the Pharmaceutical Industry

Over the past decade, there has been further insight into the number of *Talaromyces* strains isolated from mangroves that produce bioactive compounds. The pharmaceutical industry can use *Talaromyces* strains as one of the most promising “biofactories” to expand the spectrum of already accessible bioactive compounds [36]. The ability of halophilic and halotolerant fungi to produce metabolites with antioxidant activity holds considerable promise. Carotenoids are well-known pigments produced under harsh salinity conditions by halophilic microorganisms including bacteria, yeasts, algae and archaea. These pigments are rare in halophilic fungi and have only been found in one species, *Fusarium* sp. T-1 isolated from Japanese salt water. The carotenoids discovered included beta-carotene, neurosporaxanthin, neurosporaxan-thin-D-glucopyranoside, and torulene [98]. Another pigment found in high concentrations in halophilic fungi is melanin. It has not been discovered in any other halophile. In fungi, melanins exist extracellularly as polymers produced by enzymes or autooxidation in the medium, or as an independent substance within the cell walls.

Fungi of Ascomycetes and Deuteromycetes produce DHN-melanins, which are derived from 1,8-dihydroxynaphthalene (DHN). With the help of these pigments, fungi can survive under adverse environmental conditions such as intense UV radiation, extremely high temperatures and osmotic stress [99]. In addition to the ability of halophilic filamentous fungi to synthesize melanins, some of them have also been found to synthesize other pigments. For example, the species *Periconia* sp. can produce a rare blue pigment [100], and the halotolerant *A. variecolor* produces variecolorquinones (quinone compounds) with a yellow color [101]. Kumar et al. [102] reported that melanin produced by *A. bridgeri* is involved in free radical scavenging, due to its antioxidant activity. According to this study, melanin has potential applications in the pharmaceutical and cosmetic industries [102]. Exopolysaccharides (EPSs) produced by fungi have potent antioxidant effects [103]. The marine fungi [39] as well as *Aspergillus* sp. DHE6 have been found as exopolysaccharide producers [104].

The halotolerant deep fungus *A. versicolor* N2bc produces an EPS, N1, composed of manoglucogalactan with a side chain of galactofuranose units [105]; the marine fungus *A. terreus* produces a new EPS, YSS, composed mainly of mannose and galactose [106]. The coral-associated fungus *A. versicolor* LCJ-5-4 is a producer of a novel EPS-AVP composed of mannoglucan and a side chain of mannopriranose trisaccharide units [107]. The physicochemical and biological properties of microbial EPS determine their potential for applications in biomedicine and pharmaceuticals, as well as in agriculture, the food industry, biofilms, cosmetics, etc. [108]. Phenolic compounds also have antioxidant effects. The marine fungus *A. versicolor* is a producer of antioxidant secondary metabolites with freely rotating phenolic hydroxyl groups. The highest total flavonoid content was found in *Engyodontium album*, while high total phenolic content was found in *G. dankaliensis* [109].

Three phenols and the benzamide derivative methyl 4-(3,4-dihydroxybenzamido)butanoate with strong antioxidant activity were isolated from the algae-derived fungus *A. wonii* EN-48 [110]. Kristina Sepcic et al. [111] reported isolated halophilic and halotolerant fungi that produced hemolytic components [111]. Their hemolytic activity is most likely related to the production of organic molecules. Antimicrobial and hemolytic compounds can be used against bacteria and protozoa, while acetylcholinesterase inhibitors can be used against higher organisms that have a developed nervous system [111].

### 4.6. Antimicrobial Activity

Interestingly, not only microbes and their enzymes but also their production of secondary metabolites can be influenced by salt concentration. Some of the salt-tolerant fungi in sun salterns produce antifungal and antibacterial compounds [20,112], suggesting that they can play a role in influencing the microbial composition of saline environments. One of the greatest threats to public health is the increasing resistance of many pathogenic microorganisms to antibiotics. Their misuse over decades has reduced or eliminated their effectiveness [113]. This problem requires the search for new strategies to fight bacterial infections. One of the most important tasks in biomedical research is the discovery of new antibiotics. Organisms inhabiting saline habitats have also been found to synthesize new molecules for biomedical applications.

Fungi are considered effective producers of secondary metabolites with antimicrobial properties. Halophilic and halotolerant fungi have attracted attention in recent years due to their potential as a source of new antimicrobial agents with antibacterial and antifungal properties [114]. Some studies have shown that halophilic fungi produce bioactive compounds with antiviral properties [115]. Compound 3 and compound 6 with antiviral activity against the hepatitis A virus were isolated from an obligate halophilic fungus *Phialosimplex asmahalo*. In addition, compound 3 was effective against herpes simplex virus type 1 [116]. They could make a significant contribution to the development of novel antiviral drugs, particularly in the context of emerging viral diseases.

Halophilic and halotolerant marine fungi have been found to release secondary metabolites such as steroids, peptides, terpenes, alkaloids and polyketides, which have antiviral, antibacterial, antifungal, anticancer and cytotoxic effects [117]. Table 3 presents some compounds produced from salt-adapted fungi with antimicrobial activity.

Zhao et al. [118] reported the compounds isolated from marine fungi with antimicrobial and anticancer properties such as alterperylenol, anthraquinone derivatives, (11S)-1, 3, 6-trihydroxy-7-(1-hydroxyethyl)-anthracene-9,10-dione, 7-Acetyl-1, 3, 6-trihydroxyanthracene-9,10-dione and Stemphyperylenolect [118]. Antibacterial and antioxidant properties are found in *A. gracilis*, *A. penicillioides* and *A. flavus* [67]. Gonçalves et al. [117] reported four marine fungal strains (*A. affinis*, *E. cladophorae*, *Pen. lusitanum* and *T. aestuarinum*) that showed antibacterial activity [117]. *A. affinis* and *Pen. lusitanum* have been found to have antibacterial activity against *K. pneumoniae* [117,119] and *P. aeruginosa* [117]. They can be used as a source of antibacterial molecules. Ballav et al. [120] reported halophilic and halotolerant *Micromonospora* sp., *Kocuria* sp. and *Actinomycetes* sp., which produce antibacterial compounds against *Vibrio cholera*, *Staphylococcus aureus* and *Staphylococcus citreus* [120].

*A. gracilis*, *A. penicillioides* and *A. flavus* were found to have high antibacterial potential [6]. Wang et al. [43] studied an anthraquinone named 2-(dimethoxymethyl)-1-hydroxyanthracene-9,10-dione obtained from *A. versicolor* and reported its antibacterial activity against multidrug-resistant strains of *S. aureus* ATCC 43300 and some strains of *Vibrio* [43]. The halophilic strain *A. protuberus* MUT 3638, isolated from cold seawater, has significant antimicrobial activity against *S. aureus*, *K. pneumoniae*, *B. metallica* and *A. baumanii*. This fungus could grow in a wide range of salinity, pH and temperature [122].

Compounds with potent antibacterial, antifungal, and/or cytotoxic activities were obtained from fungal strains isolated from marine plants off Qingdao, China. Some of them induce membrane hyperpolarization in *C. michiganensis* without altering the integrity of the cell membrane [118]. Engyodontiumone A-J and 2-methoxyl-cordyol C, isolated from the deep-sea strain *Engyodontium album* DFFSCS02, show antibacterial activity against *E. coli* and *Bacillus subtilis* and antifungal activity (Engyodontiumin A) against *A. niger* and antibacterial activity against multi-resistant *S. aureus*, *Vibrio vulnificus*, *V. rotiferianus* and *V. campbellii* [121]. Bao et al. [123] reported alkaloids isolated from deep-sea *Arthrinium* sp. The isolated alcaloid UJNMF0008 shows antibacterial activity against *Mycobacterium smegmatis* and *S. aureus* [123]. Halophilic fungi have also been studied for their ability to produce antifungal compounds. Due to their harsh ecological niche, they have developed potent antifungal metabolites as part of their defense mechanisms. These compounds have the potential for use in agriculture, food preservation and pharmaceutical applications.

Halophilic *Penicillium* species are known to produce both antibacterial compounds and antifungal substances. The extracts of *Pen. lusitanum* mycelium were found to have an inhibitory effect on *C. albicans*, meaning they could be useful in the treatment of candidiasis [117]. Fumiquinazolines A–G produced by *A. fumigatus* inhibited the proliferation of CDC2 mutant mouse cells (tsFT210) and showed antifungal activity [124]. Halotolerant and halophilic fungi isolated from the soil of solar salt deserts were examined for antimicrobial activity. Fungal mycelium extracts and culture broths of *P. citrinum* NM-3 and five strains of *A. subalbidus* were found to possess activity against Gram-positive bacteria (*Enterococcus faecalis*, *Micrococcus luteus*, *S. aureus* and methicillin-resistant *S. aureus*) and Gram-negative bacteria (*E. coli*, *Salmonella typhi*) and antifungal activity against *C. albicans* and *A. fumigatus* [20].

It has been proven that some halophilic fungi belonging to these genera are capable of producing a range of polyketides, which include antibiotics such as penicillins, as well as extrolites and penicillic acids [125]. Given that the main global threats are cancer and antimicrobial resistance to antiviral drugs and antibiotics, halophilic fungi, which have not yet been fully studied, could be exploited as a source of secondary metabolites to help solve these problems.

## 5. Conclusions

Although halophilic fungi are promising sources of valuable bioactive compounds and have enormous potential for application in many of the areas mentioned in this review, there are several challenges in exploiting their potential. One of the largest obstacles is the limited knowledge of their ecology, diversity and biology. As polyextremophiles, elucidating their survival mechanisms would enrich our knowledge of life in extreme habitats. New knowledge about the adaptive strategies of halophilic fungi can serve as a predictive tool in exploring the limits of life. Understanding their metabolic pathways and the environmental influences that trigger the production of valuable compounds is crucial for optimizing the processes of their cultivation and extraction. Furthermore, the isolation and identification of specific compounds, as well as the assessment of their safety and efficacy, are essential steps in the process of discovering valuable metabolites with broad applications in various fields. Extremozymes produced by halophiles have a number of advantages over their conventional homologues and are of increasing interest. This outlines the essential importance of studies on the biology and potential of halophylic fungi, as well as the importance of future research on the issue.

## Figures and Tables

**Figure 1 microorganisms-13-00175-f001:**
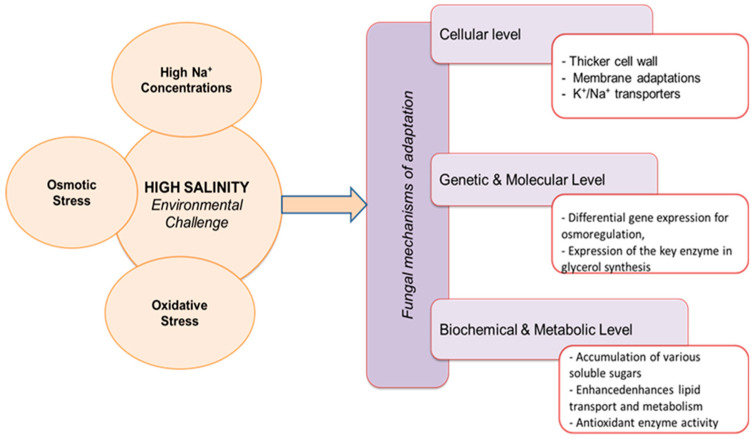
Fungal mechanisms of adaptations to high salinity.

**Table 1 microorganisms-13-00175-t001:** Some halophilic fungi and their habitats.

Halophilic Fungal Strain	Source of Isolation	References
Dead Sea	*Gymnascella marismortuispecies*, *Penicillium westlingii* and *Ulocladium chlamydosporum* *Penicillium westlingii* and *G. marismortui*	[11,12,32,33,34]
Man-made solar saltern	*Penicillium imranianum*	[16]
Solar salt marshes	*Penicillium imranianum*	[4,15,16]
Solar salt plant	*A. gracilis* TISTR 3638 *A. flavus* (TISTR 3637)	[6,35]
Estuaries and salt lakes	Genera *Aspergillus* and *Penicillium*	[14]
Mangroves	*Talaromyces* Genera *Aspergillus* and *Penicillium*	[15,36]
Saltern in the Republic of Korea	*Aspergillus reticulatus* SK1-1	[37]
Marine fungi	*Ascocratera manglicola*, *Astrosphaeriella striatispora*, *Cryptovalsa halosarceicola*, *Linocarpon bipolaris* and *Rhizophila marina*, *Eutypella* sp. MCCC *Epicoccum nigrum* JJY40, *Gymnascella dankaliensis*, *Nigrospora oryzae* and *Chaetomium globosum*	[33,38,39]
Deep-sea	*Diaporthe longicolla* FS429 *Phomopsis lithocarpus* FS508 *Penicillium fellutanum* HDN14-323 *Engyodontium album* DFFSCS02	[40,41,42,43]
Antarctic sediment	*Penicillium granulatum* MCCC 3A00475	[44]

**Table 2 microorganisms-13-00175-t002:** Fungal adaptations to high salinity.

Level of Adaptation	Adaptations	References
Cellular	Cell wall thickening and change in lipid composition	[49]
	Presence of pigments (melanins and carotenoids)	[50,51,52]
	Biosynthesis of crystalline chitin and α-glucan branching	[55]
	ENA ATPase, a plasma membrane protein, as one of the mechanisms of salt extrusion or Na^+^/K^+^ transport.	[7]
	Maintaining a low sterol to phospholipid ratio and reducing fatty acid cycle length and phospholipid saturation	[56]
	Production of hydrophobins. In addition, these proteins strengthen the cell wall and make it more rigid for halo adaptation	[45]
Genetic	A set of genes encoding stress response proteins (heat shock proteins and chaperonins) and the expression of a glycerol synthesis enzyme, Gpd1, which is under the control of the high-osmolar glycerol (HOG) signaling pathway.	[45,57]
Enzymatic and metabolic pathways	Secretion of extracellular polysaccharides that can serve as protective agents in the presence of high salt concentrations and during desiccation	[69]
	Exclusion of salts	
	Synthesizing or accumulating compatible organic solutions, such as glycerol	[3,57]
	Antioxidant enzyme activity	[60]

**Table 3 microorganisms-13-00175-t003:** Antimicrobial activity of some salt-adapted filamentous fungi.

Salt-Adapted Filamentous Fungi	Active Sybstances	Application	References
*Phialosimplex asmahalo*	Isolated compound 3 and compound 6	Antiviral activity against Hepatitis A virus, Herpes Simplex type 1	[114,115]
Halophilic and halotolerant marine fungi (non identified)	Alterperylenol, anthraquinone derivatives, (11S)-1, 3, 6-trihydroxy-7-(1-hydroxyethyl) anthracene-9,10-dione, 7-acetyl-1, 3, 6-trihydroxyanthracene-9,10-ddione, stemphyperylenolect	Antimicrobial and anticancer properties	[116,117]
*A. affinis*, *E. cladophorae*, *Pen. lusitanum*, and *T. aestuarinum*)	Antibacterial compounds	Antibacterial activity	[116]
*A. affinis*, *Pen. lusitanum*	Antibacterial compounds	Antibacterial activity (against *K. pneumoniae* and *P. aeruginosa*)	[116,118]
*Micromonospora* sp., *Kocuria* sp., and *Actinomycetes* sp.	Antibacterial compounds	Antibacterial activity (against *Vibrio cholera*, *Staphylococcus aureus*, and *Staphylococcus citreus*)	[119]
*A. versicolor*	2-(dimethoxymethyl)-1-hydroxyanthracene-9,10-dione	Antibacterial activity (against *S. aureus* ATCC 43300, *Vibrio* sp.)	[43]
*Pen. lusitanum*	Dried crude extracts from the mycelia	Inhibitory effect on *C. albicans*	[116]
*A. affinis*	Dried crude extracts from the mycelia	Inhibitory effect on *E.coli*	[116]
*A. protuberus*	Antibacterial compounds	Antimicrobial activity against *S. aureus*, *K. pneumoniae*, *B. metallica*, and *A. baumanii*	[120]
*Engyodontium album*	Engyodontiumin A	Antifungal and antibacterial activity	[121]
*Arthrinium* sp.	Different compounds including arthpyrone B, apiosporamide, isomer of apiosporamide	Antibacterial activity against *Mycobacterium smegmatis* and *S. aureus*, *E. coli*, *P. aeruginosa* and *C. albicans*	[122,123]
*A. fumigatus*	Fumiquinazolines A–G	Antifungal activity	[123]
*P. citrinum NM-3* and *A. subalbidus strains*	Antibacterial compounds	Antibacterial activity against Gram-positive and Gram-negative bacteria	[124]

## Data Availability

Data are contained within the article.

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
