# Peer review of "Halophilic Fungi—Features and Potential Applications"

_microorganisms, 2025, doi:10.3390/microorganisms13010175_

Round 1
Reviewer 1 Report (Previous Reviewer 1)
Comments and Suggestions for Authors
In the corrected version of the manuscript Halophilic Fungi – Features and Potential Applications the authors have significantly improved its quality.
I have a few comments left.
1. Since the review is devoted to filamentous fungi, in my opinion this should be reflected in the title. In my opinion it is too general.
2. The section Biodiversity and taxonomy of fungi in saline habitats should be more focused on filamentous fungi, and it would be good to provide a dendrogram reflecting biodiversity.
3. Section 3 should also be more focused on filamentous fungi. They can certainly be compared with representatives of other groups.
4. In section 5 I would like to see not only their potential for use in remediation and water purification, but also real working technologies with their use.
Author Response
Dear Reviewer, thank you for your advices and suggestions. We prepared a revised version of our manuscript following your recommendations. We sincerely hope that in its present form, our article is suitable for publication.
Best regards!
E.Krumova
Comments and Suggestions for Authors
In the corrected version of the manuscript Halophilic Fungi – Features and Potential Applications the authors have significantly improved its quality. I have a few comments left.
- Since the review is devoted to filamentous fungi, in my opinion this should be reflected in the title. In my opinion it is too general.
Replay: Thanks for the suggestion! We were also hesitant about the title, but we think that since this review includes information on both filamentous fungi and yeasts, it would be better to talk about halophilic fungi.
- The section Biodiversity and taxonomy of fungi in saline habitats should be more focused on filamentous fungi, and it would be good to provide a dendrogram reflecting biodiversity.
Replay: We agree with you that illustration of what is described in the text is necessary to make it easier to perceive. We have added a table on biodiversity which we think will make the information easier to understand.
- Section 3 should also be more focused on filamentous fungi. They can certainly be compared with representatives of other groups.
Replay: Our review focuses on both filamentous fungi and yeasts. We have added a table that synthesizes fungal adaptations to saline habitats.
- In section 5 I would like to see not only their potential for use in remediation and water purification, but also real working technologies with their use.
Replay: In the part concerning bioremediation we added literature on mycoremediation and the advantages of using fungi compared to bacteria and algae.
Reviewer 2 Report (Previous Reviewer 3)
Comments and Suggestions for Authors
The authors corrected the previously submitted MS according to reviewer comments: corrected the keywords, added the explanation between halophilic and halotolerant organisms, improved the structure of the paper, added a comparative table and scheme of the study, added information about the future prospects in the conclusion, and added recent literature in the reference list.
The quality of the MS is much better now, and I can recommend it for publication in the journal Microorganisms.
Author Response
Dear Reviewer,
Thank you for your help!
Best regards!
E. Krumova
Reviewer 3 Report (New Reviewer)
Comments and Suggestions for Authors
General comments
Given that this is a review, the authors must ensure proper referencing. However, they write very lengthy paragraphs containing key statements but no citations. For example, Lines 33-42, 50-60, and 168-182.
Although the authors have done considerable work to discuss the various applications, they weigh more into the medical applications as is seen from Sections 4,4 to 4,6, The biotechnology application sould benefit from some more details
In the section on genetic adaptations, the authors could talk abit on the possibility of genetically engineering these fungi for better output.
Specific comments
Lines 43-47: The classification of halotolerant bacteria there is unclear. Oren distinguishes between halotolerant and halophilic organisms. Please rephrase this section.
Line 105: Delete "isolates"
Line 112: Delete "isolated"
Line 116-117: Rephrase up to [22]
Line 120: ...and thus reduces its reduction...this is confusing. Please rephrase
Line 156: ...sunlight is another its adaptation... something is lacking
Line 186-195: If you mention changes at the genetic level, you should mention the genes that encode for the enzyme production. Mention the osmoactive genes.
Line 432: Delete isolated
Line 459: .......i.e. h. has antioxidant activity..... check that
Comments on the Quality of English Language
Please proofread the manuscript for minor grammatical errors
Author Response
Dear Reviewer, thank you for your advices and suggestions. We prepared a revised version of our manuscript following your recommendations. We sincerely hope that in its present form, our article is suitable for publication.
Best regards!
- Krumova
General comments
- Given that this is a review, the authors must ensure proper referencing. However, they write very lengthy paragraphs containing key statements but no citations. For example, Lines 33-42, 50-60, and 168-182.
Replay: We have made adjustments to achieve correct referencing.
- Although the authors have done considerable work to discuss the various applications, they weigh more into the medical applications as is seen from Sections 4,4 to 4,6, The biotechnology application sould benefit from some more details
Replay: We have added information on the application of halophilic fungi in biotechnology.
- In the section on genetic adaptations, the authors could talk abit on the possibility of genetically engineering these fungi for better output.
Replay: We added information in line with your recommendation on the possibilities of genetic engineering to improve some target properties of halophytic fungi.
Line 120: ...and thus reduces its reduction...this is confusing. Please rephrase
Replay: We have paraphrased the above sentence.
Line 156: ...sunlight is another its adaptation... something is lacking
Replay: We have corrected the sentence.
Line 186-195: If you mention changes at the genetic level, you should mention the genes that encode for the enzyme production. Mention the osmoactive genes.
Replay: We have added information regarding genes related to osmoregulation.
Line 432: Delete isolated
Replay: We made the correction
Line 459: .......i.e. h. has antioxidant activity..... check that
Replay: We made the correction
Round 2
Reviewer 3 Report (New Reviewer)
Comments and Suggestions for Authors
Thank you for addressing the comments and improving the manuscript. However, the added text still requires minor English language editing.
For example:
Line 328: Even in the presence of
Line 341:...play an important role
This manuscript is a resubmission of an earlier submission. The following is a list of the peer review reports and author responses from that submission.
Round 1
Reviewer 1 Report
Comments and Suggestions for Authors
The manuscript entitled “Halophilic fungi - features and potential applications» is a review devoted to filamentous fungi adapted to high-salt environments which are novel and valuable source of enzymes, biomolecules, and biomaterials. This paper also discusses detailing their adaptive mechanisms including the development of salt-tolerant enzymes and bioremediation strategies for saline habitats. To my mind this manuscript is corresponding to the aims and scopes of the “Microorganisms” journal. I am ready to recommend it for publication after correcting several comments.
1. The abstract should be more specific for review articles as well. It should show what conclusions the authors have made after studying the body of literature data. In addition, there are repetitions in the abstract that should be removed.
2. The introduction, in my opinion, should be expanded to include a more detailed description of the research problem and the purpose of this review.
3. This is the first time I've seen a review without a single table or picture. Systematization of data in the form of diagrams or tables always simplifies the perception of the material.
4. Although the authors stated that they pay a lot of attention to the mechanisms of adaptation to conditions of high salt concentrations, this topic is covered rather superficially in the text of the review. and it seems to me that there is no fundamentally new data in it.
5. I find it very strange that the authors did not pay attention to the biodiversity and taxonomy of fungi in these habitats, although the abstract says that they pay attention to filamentous fungi, in the text itself there is very little description of diversity\
6. In my opinion, the conclusion to the work is also too short and general for a review article.
general conclusion on the work. in this form in the manuscript the main attention is paid to metabolites of halophilic fungi and their application in various practical spheres, from the point of view of description of the diversity of fungi in these conditions and fundamental bases of metabolism, which I would like to see in a microbiological journal, in this manuscripts there is little information it is poorly systematized and does not represent novelty.
Comments on the Quality of English LanguageI would advise authors to ask for help from a native speaker
Reviewer 2 Report
Comments and Suggestions for Authors
The item does not represent a novelty, and even recent reviews have been already published such as:
· Agrawal, S., Chavan, P., & Dufossé, L. (2024). Hidden Treasure: Halophilic Fungi as a Repository of Bioactive Lead Compounds. Journal of Fungi, 10(4), 290.
· Hmad, I. B., & Gargouri, A. (2023). Halophilic filamentous fungi and their enzymes: Potential biotechnological applications. Journal of Biotechnology.
· Ali, I., Khaliq, S., Sajid, S., & Akbar, A. (2019). Biotechnological applications of halophilic fungi: past, present, and future. Fungi in extreme environments: Ecological role and biotechnological significance, 291-306.
The review is written discursive and generalistic without ever providing an original interpretation of diversity data, growth/optimum limits (aw), growth temperature, etc. Information is not always accompanied by the appropriate citation (e.g., L28 extensively studied examples including the Dead Sea...), no table, no figures to synthesize concepts.
For these reasons, a rejection is recommended.
Reviewer 3 Report
Comments and Suggestions for Authors
The reviewed MS is dedicated to the study of halophilic fungi application. In a recent publication biochemistry of these organisms was discussed (Ali et al., 2019; Śliżewska et al., 2022). But in the reviewed paper, the broader topic of using halophilic fungi in biotechnology is presented. The results of the MS are important for mycology and microbiology. The basic statements and conclusions of the MS are supported by analyzing reference. The paper is clear and will be of interest to readers. But for better understanding the studied topic, it is necessary to add figures and tables and make some other corrections.
Major comments:
- Please, don’t use for keywords the terms from the title.
- Explain differences between halophilic and halotolerant fungi.
- Please correct the structure of Section 3 "Applications of halophilic filamentous fungi". Change the numbers of sections 4 to 3.1, 5 to 3.2, and so on.
- Add comparative tables about using halophilic and halotolerant fungi in bioremediation and their antimicrobial activity.
- Please summarize the basic ideas of your review in one scheme.
- In "Conclusion" discuss the future prospects of halophilic fungi investigation.
- Add at least 40–50 recent publications (published within the last 5 years) to the reference list. You should cite mostly resent papers. Only about 29 publications from 91 in the reference list are new.
Minor comments:
Lines 27-40: Add reference citations.
Line 78: Correct S. cerevisiae to Saccharomyces cerevisiae. It is the first mention of this species. Here and further during the first mention of the species add the name(s) of the author(s).
Lines 168-184: Add references.
Line 218: Delete "Heavy metals".
Line 261: Add author names for the genera Aspergillus, Pencillium, and Fusarium.
Delete number 10 in the section "Conclusion".
Reference 91: Please, add the year of the publication.
Comments on the Quality of English Language
The English could be improved to more clearly express the research.